# Recent Advances in Apical Periodontitis Treatment: A Narrative Review

**DOI:** 10.3390/bioengineering10040488

**Published:** 2023-04-19

**Authors:** Zulema Arias, Mohammed Zahedul Islam Nizami, Xiaoting Chen, Xinyi Chai, Bin Xu, Canyan Kuang, Kazuhiro Omori, Shogo Takashiba

**Affiliations:** 1Department of Pathophysiology-Periodontal Science, Graduate School of Medicine, Dentistry and Pharmaceutical Sciences, Okayama University, 2-5-1 Shikata-cho, Kita-ku, Okayama 700-8558, Japan; chen_xt@s.okayama-u.ac.jp (X.C.); chaixinyi_2023@s.okayama-u.ac.jp (X.C.); phid742y@s.okayama-u.ac.jp (B.X.); masa_kuang@s.okayama-u.ac.jp (C.K.); kazu@okayama-u.ac.jp (K.O.); stakashi@okayama-u.ac.jp (S.T.); 2Restorative Dental Sciences, Faculty of Dentistry, The University of Hong Kong, Prince Philip Dental Hospital, 34 Hospital Road, Sai Ying Pun, Hong Kong SAR 999077, China; nizami01@hku.hk

**Keywords:** apical periodontitis, contemporary approaches, nonsurgical endodontic treatment, immune inflammatory disease, alternative treatments

## Abstract

Apical periodontitis is an inflammatory response caused by pulp infection. It induces bone resorption in the apical and periapical regions of the tooth. The most conservative approach to treat this condition is nonsurgical endodontic treatment. However, clinical failure has been reported with this approach; thus, alternative procedures are required. This review highlights recent literature regarding advanced approaches for the treatment of apical periodontitis. Various therapies, including biological medications, antioxidants, specialized pro-resolving lipid mediators, and stem cell therapy, have been tested to increase the success rate of treatment for apical periodontitis. Some of these approaches remain in the in vivo phase of research, while others have just entered the translational research phase to validate clinical application. However, a detailed understanding of the molecular mechanisms that occur during development of the immunoinflammatory reaction in apical periodontitis remains unclear. The aim of this review was to summarize advanced approaches for the treatment of apical periodontitis. Further research can confirm the potential of these alternative nonsurgical endodontic treatment approaches.

## 1. Introduction

Apical periodontitis (AP) is one of the most prevalent inflammatory lesions involving the jaw. A recent systematic review revealed that 52% of the worldwide population and 50% of the global adult population has at least one tooth with AP, and that the prevalence of AP is much higher among root-filled teeth (39%) than among nontreated teeth (3%) [1]. Furthermore, several conditions, including fused roots [2], short root canal fillings, and crown restorations [3]; imaging procedures, such as periapical radiography and cone-beam computed tomography (CBCT); and systemic conditions, particularly diabetes, cardiovascular disease, and smoking, influence the presence of periapical lesions, contributing to a tendency for a higher incidence of AP [1]. Another systematic review compared data from 2012 and 2020 and found a 6.3% versus 5.4% increase in the prevalence of AP in the adult population [4]. Because the jaws are in direct contact with the bone marrow through the teeth, there is no epithelial barrier that can limit the spread of the infection. Immunological responses play an important role in moderating the invasion of endodontic microbiota into other tissues [5]. The pathogenic properties of bacteria, including antigenicity, mitogenic activity, chemotaxis, and enzymatic histolysis, and the body’s defense mechanisms, such as immune cell products and intercellular messengers, play an active role in the inflammatory response, which ultimately results in the destruction of periapical tissues. This apical inflammatory infection cannot be resolved without intervention because the contents of the necrotic root are beyond the body’s defense systems [6].

Therefore, complete disinfection and subsequent three-dimensional sealing of the root canal system are essential for achieving endodontic success [6]. Unfortunately, despite protective measures involving the use of the latest antimicrobial agents, root canal preparation, and root canal filling techniques, the incidence of endodontic failure remains high [7]. Inadequate coronal sealing, poor or overstretched root canal fillings, complications during mechanical preparation, and untreated canals serve as factors that can cause endodontic failure due to the persistence of pathogenic bacteria.

The major protagonists in the highly complicated intracellular milieu of the tooth are odontoblasts. Odontoblasts are key mesenchymal components in the tooth and have polarized nuclei with cytoplasmic processes that extend from the cell body to the predentine. Odontoblasts are not standard mineralized tissue producers; this is because the cell bodies are not included in the calcified tissue, so there is no physiological remodeling and replacement of dentin. As soon as the calcified matrix starts to break down, odontoblasts are the first to encounter toxins and compounds secreted by oral bacteria; therefore, they play an active and central role as mediators of inflammation and the tissue repair process [8].

### Current Challenges in the Diagnosis and Treatment of Apical Periodontitis

Dental radiographs, which are the standard adjunct diagnostic tool for the detection of AP, have numerous limitations. Furthermore, it has been reported that small lesions can only be detected on radiographs when there is cortical plate perforation; this means that highly contaminated small periapical lesions cannot be detected [9]. Several reports have shown that magnetic resonance imaging (MRI) findings are more consistent with histological findings for AP diagnosis than CBCT findings [10]. However, these adjunct imaging tools are expensive and not feasible for use in daily clinical practice.

Given the variability and complexity of the internal anatomy of teeth, which includes C-shaped root canals, curvatures, and lateral and apical ramifications, complete disinfection of contaminated root canals is difficult because of underlying pulp necrosis [11].

Furthermore, even the most efficient chemo-mechanical procedures cannot completely eliminate the involved microorganisms because they are mainly organized in biofilms; this makes them more resistant than planktonic bacteria [12]. Therefore, the incidence of refractory AP (RAP) is high. In addition, a recent review revealed that *Enterococcus faecalis*, one of the predominant bacteria in RAP, displays several virulence mechanisms that play a role in macrophage and osteoblast responses [13].

Various clinical approaches involving nonsurgical endodontic treatment for the treatment of AP have been developed. Antimicrobial agents such as sodium hypochlorite, chlorhexidine, and calcium hydroxide are well-known, classic adjuvants that are used to eliminate resistant bacteria that invade the apical root environment. Irrigation, a key factor in the success of endodontic treatment, has several important functions depending on the irrigant used; these include destruction of the microorganisms in the root canal, dissolution of necrotic and inflamed tissue, removal of dentinal debris, prevention of bacterial extrusion into the periapical tissues, reduction of friction, improvement of the cutting effectiveness, and cooling of the endodontic file and tooth. Importantly, it affects areas of the root canal wall that are not accessible by mechanical instrumentation. Sodium hypochlorite, because of its specific ability to dissolve organic matter, is widely considered the primary irrigant of choice to effectively kill and remove bacteria and necrotic tissue remnants in the canal [14,15]. However, a major limitation with the use of these antimicrobial agents is the development of bacterial resistance. Furthermore, flare-ups are caused by the persistence of bacteria that form biofilms within the root canal; these persist over time in the most insignificant and remote spaces of the canals [16]. Over the past few years, alternative pharmacological and cell therapies to combat the immune–inflammatory response that leads to apical bone destruction have been tested. These alternative therapies include regenerative endodontics, which uses stem cells to induce revitalization of teeth with AP and stimulate apical bone regeneration in immature and nonvital adult teeth; biological medications, such as vaccines, blood components, gene therapies, and monoclonal antibodies; antioxidants, used to decrease the generation of reactive oxygen species (ROS), which are abundant during AP; probiotics, which inhibit pathogenic microbiota by competition; and specialized pro-resolving lipid mediators (SPMs), which are normally present during the phase of resolution of inflammation in AP. With the development of healthcare innovations, research on these alternative therapies for the treatment of AP has been increasing. However, to date, only a few reviews have individually discussed these treatments. The aim of this narrative review was to compile information on these new therapies, outline their advantages and limitations, bring up-to-date information for clinicians, and encourage research to increase treatment success rates for this highly prevalent oral pathology.

## 2. Contemporary Therapies

### 2.1. Regenerative Endodontics and Apical Periodontitis

Regeneration of the pulp–dentin complex has become a major challenge for clinicians and scientists after it was introduced by the American Dental Association in 2011. Although research shows that regenerative endodontics is more effective than standard root canal treatment, many previous attempts have not been successful, with only limited tissue regeneration in previously infected teeth [17].

Regeneration occurs in a sterile microenvironment, using tissue-forming processes to replace inflammatory tissues with local or ectopic tissues [18]. Control of infection due to stem cell damage and further tissue recovery processes is crucial. Appropriate sterilization is crucial for pulp tissue regeneration [19]. The degree and chronicity of past infections affect tissue healing and regeneration processes [20]. Moreover, regeneration of pulp tissue in root canal-treated teeth requires the regeneration of all types of resident cells in pulp tissue. Importantly, regenerative endodontics is not possible without revascularization, which facilitates the nourishment of cells and adjacent tissues through an adequate blood supply. In addition, age, apical diameter, and root canal disinfection may alter the prognosis of regeneration. Therefore, a better prognosis is observed for younger patients, patients with a healthy and strong immune system, and patients with larger apical diameters, which facilitate better blood flow.

Pulp revascularization involves the reconstruction of blood vessels in the root canal. Pulp regeneration involves the restoration of odontoblasts; parasympathetic, sympathetic, and nociceptive nerve fibers; fibroblasts; and stem cells [21]. Although revascularization restores blood flow to the pulp, it cannot achieve pulp regeneration alone. Development of stem cells that will eventually differentiate into fully functional odontoblasts and nerve fibers is also necessary [22]. Stem cells are classified as totipotent, pluripotent, multipotent, and unipotent. Pluripotent stem cells are embryonic cells that can differentiate into any of the three different germ layers: the endoderm, mesoderm, and ectoderm. Conversely, multipotent stem cells can differentiate into specialized cells of the tissue of origin from a single germ layer [23]. The goal of regenerative endodontics is to obtain stem cells that can differentiate into odontoblasts. To date, five types of stem cells have reportedly shown the ability to differentiate into odontoblasts (Figure 1): stem cells of human exfoliated deciduous teeth (SHEDs), dental pulp stem cells, stem cells of the apical papilla (SCAPs), bone marrow-derived mesenchymal stem cells (BMMSCs), and dental follicle progenitor cells (DFPCs) [24]. Furthermore, a scaffold, such as platelet-rich fibrin or platelet-rich plasma, is a structure used in tissue engineering that can help proliferate cells until the framework is replaced. [21,24]. In addition, growth factors regulate the proliferation and directional differentiation of stem cells (Figure 1).

#### Mechanism of Action of Tissue Engineering for the Treatment of Apical Periodontitis

Three major factors in tissue engineering play a pivotal role in in the pulp regeneration process: stem cells, scaffolds, and growth factors (Figure 1a). After lancinating the apical papilla, stem cells, together with blood, flow into the root canal to generate SCAPs, iPAPCs, PDLSCs, and BMMSCs to stimulate endogenous stem cells for pulp regeneration [17]. SHEDs, SCAPs, BMMSCs, and DFPCs can differentiate into odontoblast-like cells (Figure 1b). Biofilms integrated into dentin can activate the immune defense system and affect cell adhesion, stem cell proliferation, and differentiation. Therefore, the root canal must be fully disinfected to promote regeneration [24]. Natural scaffolds support stem cell proliferation and differentiation, while growth factors regulate proliferation and directional differentiation of stem cells [17,24,25,26]. Endogenous growth factors primarily come from dentin and blood clots in the root canal; they are released into the root canal through root canal preparation and flushing [24] (Figure 1c,d).

The use of platelet-poor plasma as the supporting matrix for DPSCs and umbilical cord mesenchymal stem cells, with growth factors such as vascular endothelial growth factor, has been reported to enhance angiogenesis and formation of new mineralized tissue [27]. Thus, it could be a viable option for regeneration. In addition, platelet-rich plasma scaffolds showed significantly better results than platelet-poor plasma scaffolds, probably because a higher number of platelets would correspond to increased secretion of growth factors [28]. However, a high concentration of platelets can impair stem cell differentiation and proliferation. Consistent with this theory, flow cytometry in a study revealed that stem cell proliferation decreased with 5% platelet-rich plasma and increased with 0.5%/1% platelet-rich plasma [29].

Several studies have analyzed the effects of the apical diameter of teeth on the results of regenerative endodontics in cases with necrotic pulp [30]. Most studies suggest that endodontic regeneration is more feasible in young immature teeth with a wider apical diameter because of the apical size and a greater number of stem cells [31,32,33]. However, there is evidence of clinical success in mature teeth treated with regenerative endodontics [34]. Investigations using scaffolds with blood clots, platelet-rich plasma, platelet-rich fibrin, and other organic or synthetic scaffolds with/without signaling molecules and stem cells demonstrated that the 3 month clinical success rate ranged from 90% to 95%, depending on the age of the patient and the etiology of the pulp necrosis. However, further research and long-term evaluation are necessary to verify these results.

### 2.2. Biologic Medications for the Treatment of Apical Periodontitis

Biologic medications, biopharmaceuticals, and biological agents are substances that are manufactured or extracted from biological sources and target specific proinflammatory cells or their products. These include vaccines, blood components, recombinant tissue therapeutic proteins, stem cells, and gene therapy.

The immune system plays an important role in the success of nonsurgical endodontic treatment. In some immunocompromised patients, such as those with diabetes, the resolution of AP is delayed. In fact, among immunocompromised patients, there is an increase in the prevalence of periodontal disease in teeth with endodontic infection and a decrease in the success of endodontic treatment in teeth with preoperative AP [35,36]. Furthermore, genetic factors that control the host immune response have also been shown to influence the resolution of permanent chronic AP. For example, genetic markers associated with increased interleukin (IL)-1β production increase susceptibility to persistent AP [37]. It has also been concluded that genetic polymorphisms in RANK and RANKL are associated with recurrent AP [38]. All these studies concluded that the immune system plays a key role in the outcome of nonsurgical endodontic treatment. It is well established that inflammation induced by microbes and tissue injury is a nonspecific or innate immune response that is activated as part of the first defense mechanism under normal conditions. However, when certain diseases impair inflammation, the immune response can persist, leading to chronic inflammation [39].

Recently, drugs developed by genetic engineering (recombinant biotechnology), also known as biologics or biological drugs, were developed. These include humanized or chimeric (mouse–human) monoclonal antibodies or modified fusion proteins that target specific proteins to block critical steps in the inflammatory process [40]. These drugs can be categorized into three main classes: tumor necrosis factor-α (TNF-α) inhibitors, lymphocyte modulators (T-cell and B-cell modulators), and IL modulators [41].

#### 2.2.1. Mechanism of Action of Biological Medications for the Treatment of Apical Periodontitis

Bacterial invasion stimulates periodontal cells and immunocytes to secrete proinflammatory cytokines (IL-1, IL-6, and TNF-α). These cytokines induce the activation and recruitment of specific immune cell subsets, which, along with their products (such as cytokines), play key roles in the development of AP [42,43]. Biological agents such as monoclonal antibodies and fusion proteins can directly reduce the activity of specific targets, such as immunocytes and inflammatory mediators [44]; this is a potential approach for the treatment of AP. To date, the following biological medications have been tested as treatment approaches for AP: TNF-α and cathepsin K inhibitors. In addition, regulatory T-cells (Tregs), one of the CD4(+) T-cell subpopulations, act as protectors during the inflammatory reaction. Accordingly, Treg therapy has also been applied to the treatment of AP [42]. The mechanism of action of biological drugs is described in the following sections.

#### 2.2.2. TNF-α Inhibitors

TNF-α is a key proinflammatory cytokine that plays a vital role in immunologically driven diseases. It stimulates leukocyte activation and recruitment to the inflammatory site and is a ligand for a specific membrane receptor (the TNF receptor). The binding of TNF-α to its receptor induces inflammation or apoptosis (programmed cell death). There are two TNF-α receptors: TNF-α receptor 1 (receptor type 1, CD 120a and p55/60) and TNF-α receptor 2 (receptor type 2, CD120b and p75/80) [45].

Several studies involving periodontal research have investigated the use of TNF-α blockers for the treatment of periodontal disease. These studies report a decrease in periodontal indices [46], inflammatory cell recruitment, and osteoclast formation [47]. Conversely, only one report has described the use of adalimumab, an anti-TNF-α monoclonal antibody, for the treatment of AP in a ferret model. The study concluded that systemic administration of these biological medications did not interfere with the healing of AP. However, local administration as an intracanal medication accelerated the healing of chronic periapical lesions. Furthermore, the study concluded that adalimumab blocked the binding of TNF-α receptors and TNF-α, thus inhibiting the production of TNF-α and IL-6 [48] (Figure 2a).

Cathepsin K is a protease involved in bone remodeling and resorption that is expressed in osteoclasts. It has been demonstrated that cathepsins are indispensable for the expression of toll-like receptor 9 (TLR9), which plays an important role in the recognition of microbial products, in dendritic cells [49]. Several hypotheses have been postulated to explain the relationship between TLR9 and cathepsin K. Some experiments have shown that cathepsin K affects protein degradation, which inhibits the interaction between CpG and TLR9, and that cathepsin K-mediated proteolysis could influence configurational changes in TLR9, thereby decreasing TLR9 signal transduction [50]. Various synthetic cathepsin K inhibitors have been developed and used in animal models of AP. One of these inhibitors, NC-2300, was orally administered to rats with experimentally induced apical lesions. The size of the periapical lesion, number of tartrate-resistant acid phosphatase (TRAP)-positive osteoclasts, and number of major histocompatibility complex class II (MHC II) molecule-expressing macrophages decreased significantly in the treated group when compared to those in the control group (Figure 2(b-1)) [51]. Another cathepsin K inhibitor, odanacatib, has been tested in mouse models of AP and is reported to reduce bone resorption by inhibiting osteoclast activation and differentiation [52] (Figure 2(b-2)).

Another group of researchers performed gene therapy using AAV2 as a vector to knockdown cathepsin K. They confirmed that destruction of the periapical region was reduced after administration of AAV2 in the root canal in a mouse model of periapical lesions. This approach resulted in a decrease in infiltrating T-cells and lowered the expression of inflammatory mediators. Furthermore, it has been reported that direct targeting of osteoclasts by AAV2 could be more specific and effective in reducing inflammatory bone resorption than targeting of inflammatory mediators for an indirect decrease in bone resorption. In addition, osteoclast and macrophage markers (i.e., Acp5 and CD115, as well as IL-1α, IL-1β, and IL-17α) were also decreased in periapical lesions; accordingly, the study concluded that AAV2 affects osteoclast function [53] (Figure 2(b-3)).

Recent studies [54] have found that local silencing of Atp6v1c1, a subunit expressed during osteoclast differentiation, which stimulates the release of cathepsin K, can reduce bone resorption by impairing osteoclast activation and reducing the expression of proinflammatory cytokines (i.e., TNF-α, IL1-α, IL-13, IL-12, and IL-6) (Figure 2(b-4)). Atp6v1c1 stimulates the immune response through TLR9 and increases proton pump activity, which promotes bone resorption. In addition, there is a decrease in the number of CD3-positive T-cells and inflammatory cytokines. It has also been demonstrated that AAV-sh-Atp6v1c1 protects periodontal ligament cells from destruction. The study concluded that silencing Atp6v1c1 could be used as an alternative treatment for AP with osteoclast overactivation [54]. However, this mechanism requires further investigation.

#### 2.2.3. Treg Therapy

Tregs are a subtype of helper T-cell lymphocytes (CD4T). They work to maintain immune homeostasis and self-tolerance and are characterized by FOXP3 expression. In localized inflammation, such as that in AP, IL-6 stimulates differentiation of Tregs to T-helper (Th)17 cells that secrete IL-17, a proinflammatory cytokine. Many studies have found that Tregs reduce inflammatory overreactions during periapical inflammation; thus, they are correlated with the attenuation of periapical lesion progression [55,56]. CCL22 is a chemotactic cytokine that stimulates Treg migration. In one study, after injection of CCL22 into the root canal in an experimental mouse model, Tregs were transferred and restricted to apical lesions; this coincided with the downregulation of Th1 and Th17 (proinflammatory cells) and the pro-healing phase [56] (Figure 2c).

### 2.3. Antioxidants for the Treatment of Apical Periodontitis

ROS and antioxidant mechanisms balance each other to maintain normal physiological processes. Oxidative stress is defined as disruption of the pro-oxidant–antioxidant balance in favor of the former, resulting in disruption of redox signaling and/or molecular damage [57].

The production of ROS by phagocytes in response to bacterial attack is an important host defense mechanism; however, disturbances in redox homeostasis can lead to tissue damage. During endodontic infection, activation of TLR ligation on the phagocytic cell surface triggers phagocytosis, synthesis of ROS, activation of humoral and cellular responses, and production of inflammatory mediators. Increased ROS disrupts normal redox homeostasis and places cells in a state of oxidative stress, resulting in molecular damage and disturbed redox signaling, loss of bone homeostasis, increased proinflammatory mediators, overexpression of matrix metalloproteinase (MMP), and apical tissue breakdown [58]. Thus, ROS products play a pivotal role in the pathogenesis of AP.

#### 2.3.1. Mechanism of Action of Antioxidants for the Treatment of Apical Periodontitis

When macrophages and neutrophils phagocytize invasive pathogens, such as *E. faecalis*, phagolysosomes produce numerous oxidizing molecules to eliminate the phagocytosed microorganisms (Figure 3a) [59]. The periapical tissue is exposed to high levels of ROS; this may cause serious collateral damage secondary to molecular damage and the interruption of cell signaling, eventually inducing an inflammatory response and destruction of periapical tissue (Figure 3b) [59,60]. In patients with AP, the lower concentration of antioxidants and excessive production of ROS may lead to a pro-oxidant status, resulting in a certain degree of oxidative damage, known as oxidative stress (Figure 3c) [59]. Endogenous antioxidants include a nonenzymatic defense system (Figure 3d), an enzymatic defense system (Figure 3e), and a repair system (Figure 3f). The first two defense systems work together to play a role in prevention and detoxification, maintaining the oxidative balance in the body. The third system, the repair system, can repair damaged biomolecules.In addition to endogenous antioxidants, exogenous antioxidants can be provided via supplements (Figure 3g) to directly maintain or reconstruct redox homeostasis and offer potential preventive and therapeutic efficacy in AP. Superoxide ions produced during the removal of pathogens by phagocytes can be converted into less reactive hydrogen peroxide (H_2_O_2_) under the action of superoxide dismutase (Figure 3h) and catalase (Figure 3i), which catalyze the conversion of H_2_O_2_ into water and oxygen (Figure 3j) [61].

The following sections discuss studies on the use of antioxidants for the treatment of AP.

#### 2.3.2. NLRP3 Inflammasome Inhibitor

Lipoteichoic acid (LTA) is a potent virulence factor for *E. faecalis,* which is the most prevalent bacterium in RAP and promotes the expression of ROS through activation of the NLRP3 inflammasome [62]. Dioscin is an antioxidative stress agent with antibacterial, anti-inflammatory, and hypolipidemic effects. It blocks the nuclear translocation of nuclear factor kappa B (NF-κB), stimulates osteoclast proliferation, inhibits ROS expression, and promotes osteoblast proliferation and differentiation [63]. An in vitro study revealed that Dioscin administration to mouse macrophages inhibits LTA, blocks nuclear translocation of NF-κB, inhibits ROS expression induced by *E. faecalis*, and stimulates osteogenesis by increasing the production of alkaline phosphatase, runt-related transcription factor 2, and osteocalcin. Thus, it has been suggested that Dioscin can be used for intracanal irrigation or as a root canal sealant in teeth with RAP [64].

#### 2.3.3. Alpha-Lipoic Acid

Alpha-lipoic acid (ALA) belongs to the naturally occurring sulfhydryl family and has been proven to have antioxidant and anti-inflammatory effects [65,66]. ALA has been used to treat chronic diseases, including Alzheimer’s disease, hepatitis, diabetes mellitus, cardiovascular disease, and cancer. An animal study was performed on the basis of the hypothesis that ALA would contribute to the balance between oxidation and oxidative stress; it was found that ALA could decrease TNF-α, IL1-β, MMP-1, and MMP-2 levels in the serum of the experimental group, and that it restricted periapical bone lesions [67]. However, the study did not investigate oxidative stress markers in the blood; therefore, it could not elucidate whether the protective mechanism of ALA in AP lesions was due to its anti-inflammatory effects or its antioxidant effects.

#### 2.3.4. Selenium

Selenium is the main component of selenocysteine, which is an amino acid that plays a role in biological antioxidant defense and cell differentiation and division; it is also involved in maintenance of the immune response [68]. In one study, selenium was used as an intracanal dressing for the treatment of AP in Wistar rats; a calcium hydroxide dressing was used as a control. Selenium was found to be effective as an intracanal medication because it enhanced the immune system, allowing it to react favorably in the treatment of chronic apical lesions [69].

### 2.4. Probiotics for the Treatment of Apical Periodontitis

The concept of consuming living microorganisms as food for health benefits is growing. Probiotics contribute to a rapidly growing multimillion dollar industry, with increasing research on their ability to prevent and/or treat diseases [70]. There is evidence of a relation between chronic inflammatory diseases and dysbiosis of the gut microbiota; moreover, there is increasing interest in the interaction between the gut microbiota and the brain. Thus, probiotics are now being used as treatments for depressive disorders [71].

Dental diseases are a result of changes in the normal and stable oral microbiota. Therefore, a balance in the microbiota is very important. The oral cavity is no exception, because it can be used for preventive measures or treatments. Various studies have shown positive results with the use of probiotics, including decreases in periodontal pathogens, gingival bleeding, proinflammatory markers in the saliva or crevicular fluid, and periodontal pocket depth [72,73,74,75].

In one study, rat models of AP received supplementation of the probiotics *Lactobacillus rhamnosus* and *L. acidophilus*. The results showed that the group that received probiotics showed a significant reduction in the inflammatory process involved in AP. In addition, probiotics were shown to modulate IL levels [76]. A similar study was performed to evaluate whether the development of AP could be altered by the administration of a probiotic multistrain formula in a rat model. The probiotics administered to the rats included *L. acidophilus*, *L. salivarius*, *L. plantarum*, *L. rhamnosus*, *Bifidobacterium bifidum*, *B. animalis subs* Lactis, and *Streptococcus thermophilus*. Histological analysis showed that the control group with no probiotic administration had a more intense inflammatory infiltration than the treatment group. In addition, probiotics modulated inflammation and bone resorption in AP [77]. Another study showed a positive impact on the development and reduction of inflammation and bone resorption in rats with AP [78].

Because of the limited number of studies on the use of antioxidants for the treatment of AP, the therapeutic effect of some antioxidants on periapical inflammation remains unclear. Differences in the types of antioxidants and experimental models used may have affected the results. Therefore, further studies should determine whether antioxidants have a positive effect on AP.

### 2.5. Specialized Proresolving Lipid Mediators (SPMs) and Apical Periodontitis

As described above, classical therapeutic agents for periodontitis include anti-inflammatory drugs and immunosuppressive agents. However, blockade of inflammation refers to interference with the resolution phase of inflammation.

The three stages of inflammation (the inflammatory response, repair and regeneration, and remodeling and maturation) are essential for the appropriate progression of the inflammatory response. The remodeling and maturation phase is important for SPMs because it resolves inflammation. This phase begins 2 to 3 weeks after injury, and it can last for months or years as a result of incomplete tissue regeneration. Characteristics of this phase are the regeneration of blood vessels, resolution of inflammation, and the transformation of granulation tissue into scar tissue [79]. For many years, the resolution phase was considered to be a passive in vivo process that should disappear with time as a result of chemotactic stimulation at the site of inflammation. However, in the early 2000s, a new concept for the resolution of inflammation was introduced [80]. It was revealed that a variety of biochemical circuits were involved in the active resolution phase of inflammation, with the production of local mediators called SPMs [80]. SPMs are anti-inflammatory agonists involved in the recognition and phagocytosis of apoptotic neutrophils via macrophages at the site of inflammation [81]. They are derivatives of omega-3 essential fatty acids and belong to the eicosapentaenoic acid (EPA) and docosahexaenoic acid (DHA) series [82]. Resolvins (RVs) belong to the SPM family of proteins and are classed as resolvin E (RVE), derived from EPA, and resolvin D (RVD), derived from DHA. There are five types of RVD (RVD1–5) and three types of RVE (RVE13) [83]. One of the most interesting features of these SPMs is the ability to cooperate with catabasis, which is the return of damaged tissue to a state of homeostasis. However, the mechanism by which these molecules can shorten the time interval between maximum neutrophil infiltration and a decrease to 50% of the maximum neutrophil infiltration, thus increasing the rate of resolution, remains unknown [84].

#### 2.5.1. Mechanism of Action of SPMs for the Treatment of Apical Periodontitis

The mechanism of action of SPMs is primarily through the activation and binding of specific G-protein coupled receptors (GPCRs) in a tissue-dependent manner [85]. GPCRs are integral membrane proteins that translate extracellular signals into intracellular responses. ChemR23 and BLT1 are GPCRs; ChemR23 is mainly expressed in brain cells, dendritic cells, epithelial cells, and the kidneys, whereas BLT1 is expressed in neutrophils, eosinophils, monocytes, macrophages, and other cells [86]. RVE1 specifically binds to ChemR23 and BLT1 to enhance phagocytosis by macrophages and promote neutrophil apoptosis. It also reacts with ChemR23 to inhibit the activation of NF-κB, thus downregulating the inflammatory response, and interacts with BLT1 to attenuate the pro-inflammatory mediator leukotriene B4 (LTB4) [87]. GPR18 is also a GPCR that is primarily expressed in human leukocytes, including polymorphonuclear neutrophils, monocytes, and macrophages. The specific binding of RVD2 to GPR18 can stimulate intracellular cyclic adenosine phosphatase, thereby enhancing phagocytosis of apoptotic polymorphonuclear neutrophils [88] (Figure 4).

The inflammatory process in periapical tissue is primarily caused by the infiltration of bacterial substances from infected teeth. In the absence of treatment, persistent bacterial infection causes chronic inflammation, eventually promoting the destruction of periapical tissue (bone resorption) [43,89]. With successful root canal treatment, SPMs, including RVs, protectins, maresins, and lipoxins, may play a key role in the resolution of acute inflammation, which occurs after the vascular and cellular phases [89,90]. Bacterial invasion activates the migration and chemotaxis of polymorphonuclear neutrophils to the site of injury by inducing chemotactic signals involving LTB4 [90,91]. When neutrophils accumulate in pus or purulent exudates, they participate in tissue destruction. Simultaneously, cell–cell interactions mediate the production of lipoxins (LXs), RVs, and other SPMs [90,91]. These mediators are derived from polyunsaturated fatty acids and may also be produced by resolving macrophages and apoptotic neutrophils [87,88]. A decrease in neutrophil clearance can promote the accumulation of activated neutrophils at the inflammatory site and is a key factor in promoting tissue destruction [2,5]. Therefore, controlling neutrophil infiltration and enhancing the clearance of neutrophils at inflammatory sites by phagocytes (such as monocytes and macrophages) may reduce tissue damage and limit excessive inflammation [2]. LXs can stimulate the recruitment of nonphlogistic monocytes, prolong the survival of macrophages, and promote the clearance and phagocytosis of apoptotic neutrophils [3,8,9]. RVs can inhibit migration, infiltration, and accumulation of polymorphonuclear neutrophils [10,11]. RVs, maresins, and protectins promote phagocytosis of apoptotic neutrophils and cell debris by macrophages [3]. In addition to innate immune cells, SPMs also play an active role in the regulation of T-cells, B-cells, periodontal ligament fibroblasts (PDLFs), and genetic fibroblasts by inhibiting T-cell activity and antibody production, enhancing PDLF proliferation, and promoting wound healing [92].

In conclusion, SPMs can not only block chronic inflammation by limiting excessive inflammatory responses (such as by inhibiting polymorphonuclear neutrophil migration and infiltration and promoting macrophage phagocytosis of neutrophils), but also promote bacterial clearance and wound healing, decompose exudates, and reduce tissue damage, pain, and other collateral damage [3,4,7], all of which are crucial for preventing the progression of periapical inflammation (Figure 4). We will now discuss studies that used omega lipid acids and RVs for the treatment of AP.

#### 2.5.2. Resolvins and Apical Periodontitis

RVE1 has been used as an intracanal medication in the immature tooth of an experimental rat model in which AP was induced. The authors compared triple-medicated antibiotics (TMA) and RVE1 as intracanal medications and found that both agents reduced apical inflammation and promoted root formation; however, RVE1 reduced inflammation faster than TMA was able to [93]. In another study using RVD2 as an intracanal medication in a rat model of AP, it was found that application of RVD2 reduced inflammatory cell infiltration and promoted healing of AP with apical bone regeneration. In addition, in the RVD2 group, expressions of GPR18 (RVD2 receptor) and dentin matrix acidic phosphoprotein 1 were increased and mineralization was enhanced in the apical regions [94]. Another study investigated the expression of RVD2 and RVE1 in teeth with necrotic pulp and AP which were treated with intracanal administration of calcium hydroxide, 2% chlorhexidine gel, or N-acetylcysteine. The results showed that calcium hydroxide did not increase RV levels. In contrast, N-acetylcysteine caused a significant increase in RV levels after 14 days of intracanal application [95].

#### 2.5.3. Omega 3 Fatty Acids and Apical Periodontitis

Dietary habits are known to directly affect the healing of injury-induced inflammation. Therefore, the resolution of inflammation appears to be affected by signaling mediated by molecular components that originate from dietary intake. These molecular components are of two types. One class comprises hormones, including eicosanoids (prostaglandins and leukotrienes), which initiate inflammation, and SPMs (RVs, protectins, and maresins), which regulate the resolution phase of inflammation [96]. The other type includes gene modulators, which comprise NF-κB, a transcription factor involved in the initiation of inflammation, and 5′-adenosine monophosphate-activated protein kinase (AMPK), a factor that reduces the expression of NF-κB and plays an important role in the repair process. In a homeostatic state, all these components are in balance; however, if even one of these signaling systems is upregulated or downregulated, a state of catabasis will be established. Thus, a balanced diet is required to maintain equilibrium and homeostasis [97]. In order to promote the healing of an inflammation-induced injury, the intake of nutrients that can activate AMPK and optimize SPM production to resolve persistent inflammation should be increased, whereas the dietary intake of nutrients that promote NF-κB activity should be decreased [97]. However, dietary intake of the omega-6 fatty acid arachidonic acid should also be decreased; this is because eicosanoid fatty acids derived from omega-6 fatty acids are proinflammatory in nature and can intensify acute inflammation. An inflammasome is an intracellular structure formed in response to a microbiological pathogen-associated molecular pattern or a danger-associated molecular pattern. The increased omega-3 fatty acid levels in an anti-inflammatory diet can also reduce activation of the inflammasome response. The result of inflammasome activation is an increase in the levels of proinflammatory cytokines IL-1β and IL-18. Both DHA and EPA inhibit inflammasome activation [98].

A study conducted in a rat model of AP found that inflammatory cell infiltration and the number of TRAP-positive osteoclasts in apical lesions were significantly lower in the group fed an omega-3 fatty acid-rich diet than in the control group. Furthermore, the number of osteocalcin-positive osteoblasts significantly increased in AP treated with omega-3 fatty acids [99]. Another study conducted using a similar experimental model measured the triglyceride and cholesterol levels and periapical bone resorption in rats with prophylactic and therapeutic supplementation with omega-3 fatty acids and found that the presence of multiple AP foci increased systemic triglyceride levels. However, rats with AP that were fed omega-3 fatty acids showed a reduction in triglyceride and cholesterol levels and bone resorption around the affected apical area [100]. Recently, inflammatory cytokines, such as TNF-α, IL-6, and IL-17, as well as the number of inflammatory cells such as neutrophils, monocytes, and lymphocytes, were measured in the same experimental group. It was found that the number of leukocytes and lymphocytes and IL-6 expression decreased in the group fed on an omega-3 fatty acid-rich diet than in the group fed on a diet without omega-3 fatty acids [101].

## 3. Discussion

Some challenges in AP treatment are related to a proper diagnosis, proper control of all contributing factors, and long-term maintenance of the periodontium. Therefore, these challenges should be addressed by motivating patients, controlling risk factors, and, most importantly, by choosing the right treatment approaches [102]. It should be noted that nonsurgical AP treatment has limitations. Long-term follow-up of apical lesions and the skill of clinicians are among the most crucial limitations [103]. Therefore, advanced research is warranted to overcome these limitations and establish the ultimate treatment approach for AP.

A summary of recent advances in AP treatment is presented in Table 1. Most studies involved animal experiments, and further clinical studies are needed to translate novel findings into clinical settings. Treatment failure is an upsetting issue for both clinicians and patients. Simultaneously, surgical treatment has limitations with respect to patient compliance. Therefore, nonsurgical treatment is always preferred. Successful nonsurgical treatment of apical lesions will be a significant achievement in the field of dental treatments. Further in vivo and clinical studies are expected to overcome the associated limitations and facilitate the optimization of treatment approaches in clinical settings.

## 4. Conclusions

The worldwide prevalence of AP is high, and its treatment remains challenging. Both surgical and nonsurgical treatments have shown varying degrees of success; however, complete success has not been achieved. The findings from this review are not sufficient to conclude that modern nonsurgical treatments are the best treatment options for AP, although it can be established that they are better alternatives. This review will motivate researchers and clinicians to bridge the gap by conducting more advanced translational and clinical research that is necessary to understand the feasibility of each treatment and its mechanism of action, clinical application, and clinical success in the treatment and post-treatment prognosis of AP.

## Figures and Tables

**Figure 1 bioengineering-10-00488-f001:**
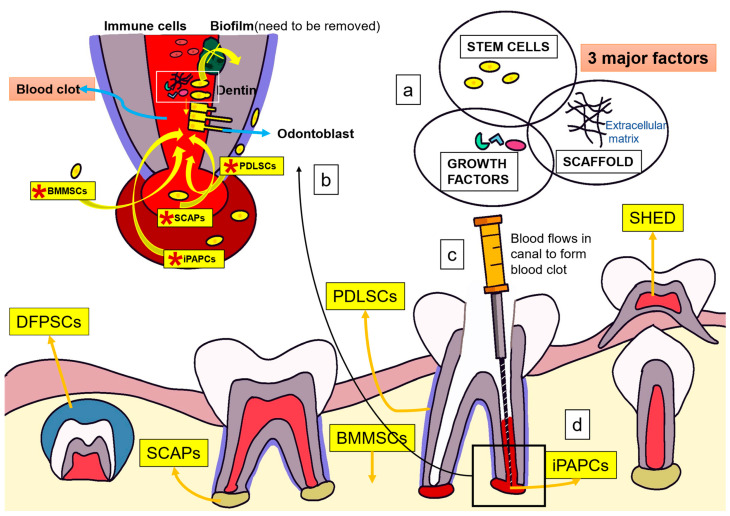
Mechanism of action of regenerative endodontics for the treatment of apical periodontitis (**a**) Three major factors in tissue engineering: stem cells, scaffolds, and growth factors. (**b**) Several stem cells including *SCAPs and *BMMSCs can differentiate into odontoblast-like cells. Biofilms integrated into dentin can activate the immune defense system and significantly affect the process of cell adhesion, and stem cell proliferation and differentiation. Root canal must be fully disinfected to promote regeneration. (**c**,**d**) Endogenous growth factors mainly come from dentin and blood clot in root canal, which can be released into root canal through root canal preparation and flushing. *SCAPs, stem cells of the apical papilla; *iPAPCs, inflamed periapical progenitor cells; *PDLSCs, periodontal ligament stem cells; *BMMSCs, bone marrow mesenchymal stem cells; SHEDs, human deciduous tooth stem cells; DPSCs, dental pulp stem cells; DFSCs, dental follicle stem cells; DFPCs, dental follicle progenitor cells.

**Figure 2 bioengineering-10-00488-f002:**
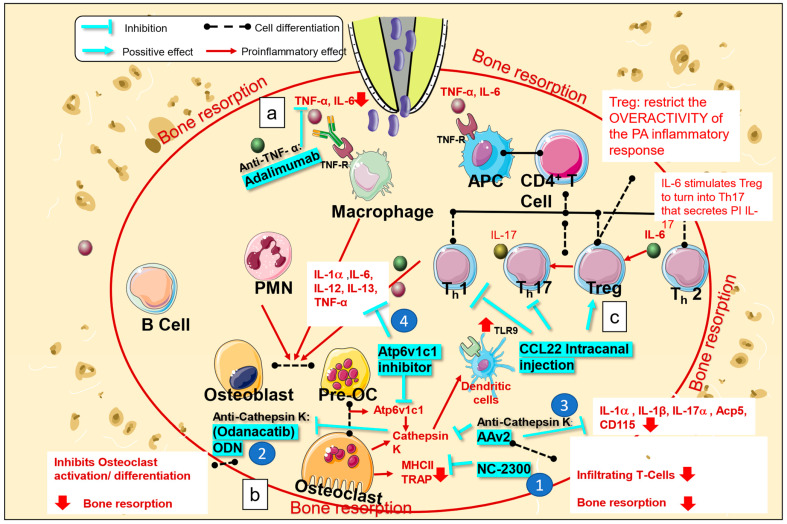
Mechanism of action of biological medications used for the treatment of apical periodontitis. Bacterial invasion in the root canal stimulates periodontal cells and immunocytes to secrete proinflammatory cytokines (interleukin IL)-1, IL-6, tumor necrosis factor (TNF-α). (**a**) Biological medications, such as anti-TNF-α monoclonal antibodies (adalimumab), block the binding of TNF-α to TNF-α receptors, subsequently decreasing TNF-α and IL-6 production. (**b-1**) NC-2300, a cathepsin K inhibitor, inhibits the expression of TRAP and MHCII in osteoclasts. (**b-2**) Odanacatib ameliorates osteoclast activation and differentiation. (**b-3**) AAv2 reduces the infiltration of T-cells and expression of osteoclasts and macrophages markers, such as Acp5, CD115, IL-1α, IL-1β, and IL-17a. (**b-4**) Atp6v1c1 inhibitors reduce bone resorption by impairing osteoclast activation and downregulating the production of TNF-α, IL1-α, IL-13, IL-12, and IL-6. (**c**) CCL22, a chemotactic cytokine, was found to stimulate Treg migration and downregulate Th1 and Th17 expression. APC, antigen presenting cell; Treg, regulatory T-cell.

**Figure 3 bioengineering-10-00488-f003:**
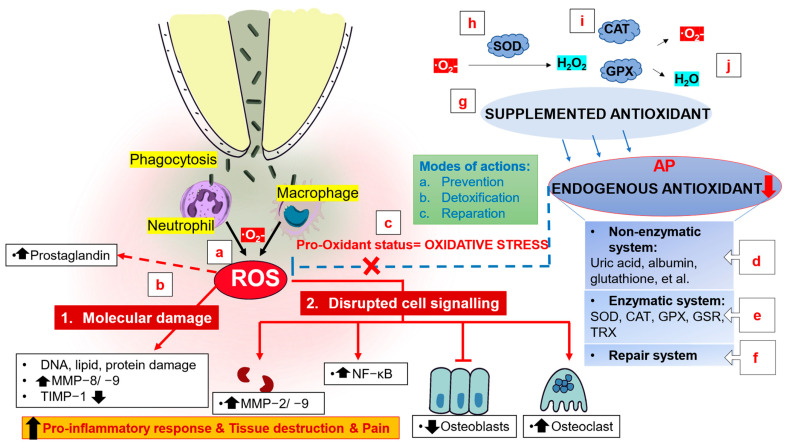
Mechanism of action of antioxidants for the treatment of apical periodontitis. (**a**) Phagolysosomes produce numerous oxidizing molecules to eliminate the phagocytosed microorganisms. (**b**) High levels of ROS in the periapical tissue may cause serious collateral damage secondary to molecular damage and the interruption of cell signaling, promoting the inflammatory response and destruction and pain of periapical tissue. (**c**) Pro-oxidant status results in a certain degree of oxidative damage which is known as oxidative stress. (**d**–**f**) Endogenous antioxidants, including a nonenzymatic defense system, an enzymatic defense system, and a repair system, play a role in prevention, detoxification, and reparation to maintain the oxidative balance and repair damaged biomolecules. (**g**) Exogenous antioxidants can be provided via supplements. Superoxide ions can be converted into less reactive hydrogen peroxide (H_2_O_2_) under the action of (**h**) superoxide dismutase, (**i**) catalase and glutathione peroxidase, which catalyze the conversion of H_2_O_2_ into (**j**) water and oxygen. SOD, superoxide dismutase; CAT, catalase; GPX, glutathione peroxidase; GSR, glutathione reductase; TRX, thioredoxin; MMP, matrix metalloproteinase; TIMP, tissue inhibitor of metalloproteinase.

**Figure 4 bioengineering-10-00488-f004:**
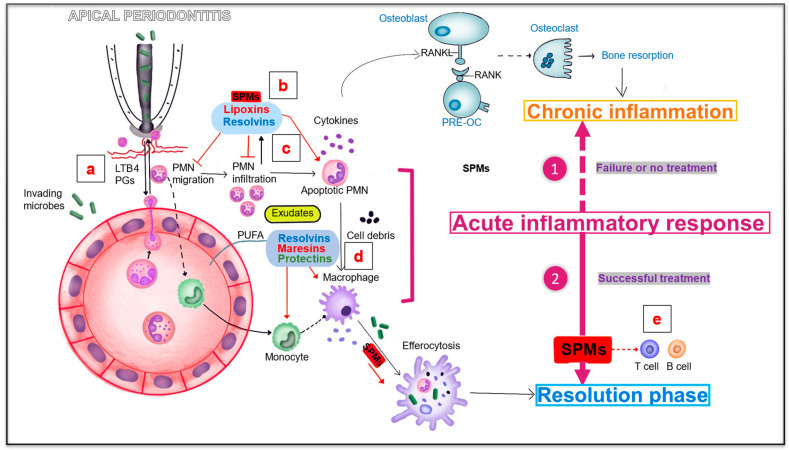
Specialized pro-resolving lipid mediators involved in apical periodontitis. (**a**) Bacterial invasion activates the migration and chemotaxis of polymorphonuclear neutrophils (PMNs) to the injury site by inducing chemotactic signals including LTB4. (**b**,**c**) SPMs including lipoxins and resolvins are derived from PUFA and may also be produced by resolving macrophages and apoptotic neutrophils. Lipotoxins can promote the clearance and phagocytosis of apoptotic neutrophils. Resolvins can inhibit the migration, infiltration and accumulation of PMN. (**d**) Resolvins, maresins and protectins can promote the phagocytosis of apoptotic neutrophils and cell debris by macrophages. (**e**) SPMS can also play an active role on T cells and B cells by inhibiting T cell activity and inhibiting antibody production. Pre-OC, osteoclast precursor; PMN, polymorphonuclear neutrophil; LTB4, leukotriene B4; PGs, prostaglandins; SPMs, specialized pro-resolving mediators; PUFA, polyunsaturated fatty acid.

**Table 1 bioengineering-10-00488-t001:** Recent advances in the treatment of apical periodontitis.

Treatment	Key Treatment Strategies	Study Subjects	References (Available Articles)
Regenerative endodontics	Scaffolds with blood-clot; platelet-rich plasma; platelet-rich fibrin Apical papilla stem cell stimulation (root canal application)	Human Dogs	[30] [55]
Adalimumab	Anti-tumor necrosis factor alpha (local and systemic administration)	Ferret Human	[48,104]
NC-2300	Cathepsin K inhibitor (oral administration)	Rats	[51]
Odanacatib	Cathepsin K inhibitor	Mice	[52]
AAV2	Cathepsin K knockdown vector (root canal application)	Mice	[53]
AAV-sh-Atp6v1c1	Recombinant adeno-associated virus-mediated Atp6v1c1	Mice	[54]
Treg-cell therapy CCL22	Targeting Treg cells Th1 & Th17 downregulation (root canal application)	Mice	[56]
Alpha-lipoic acid	Antioxidant (intraperitoneal injection)	Rats	[67]
Selenium	Antioxidant (root canal application)	Rats	[69]
Microbes	Probiotics (systemic administration)	Rats	[76]
Microbes	Probiotics (systemic administration)	Rats	[77]
Microbes	Probiotics (systemic administration)	Rats	[78]
Omega-3 fatty acid	Resolution of inflammation (systemic administration)	Rats	[99,100,101]
RVE1	Resolution of inflammation (intracanal application)	Rats	[93]
RVD2	Resolution of inflammation (intracanal application)	Rats	[94]

## Data Availability

Not applicable.

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
