# Peer review of "Recent Advances in Apical Periodontitis Treatment: A Narrative Review"

_bioengineering, 2023, doi:10.3390/bioengineering10040488_

Round 1

Reviewer 1 Report

Dear authors, I would like to take the present opportunity to congratulate you for the efforts performed while conducting the present study.

Here goes a few concerns:

I would suggest the authors to reach the 5 keyword in order to increase the traceability and visibility of the manuscript.

There is a lack of aim and rationale for the present review in the Introduction. Why do we need this review? What are the gaps on the current literature that make this review prone to be published? What is the objective of this review? These questions should be available in the last paragraph of the Introduction.

I suggest the authors to avoid abbreviations (such as AP) in the subheadings.

I believe the role of irrigation is not receiving the deserved credit on this revision. The authors state that that is a classic adjuvants, but they are classic for a reason. At least a small paragraph mentioning the role of the irrigants would be a most.

I fell that one other issue is missing here, which is the periapical lesions prevalence. This might be a good approach for the study rationale since it basically affects the population of the whole world. Another issues that is missing may be the clinical conditions that may be more associated to it. In these cases I suggest the studies DOI 10.1016/j.joen.2020.03.003 and 10.1111/iej.13256 to support the previous recommendation.

I recommend the authors to expose a few limitations of current knowledge and further studies considerations in a last subheading previous to the Conclusions section.

Reviewer 2 Report

The reviewer really appreciates the efforts of the authors to conduct this study which has good clinical significance. The manuscript is well written without leaving major issues in it. The illustrations are self-explanatory.  However, the reviewer thinks a minor adjustment in the logical sequence will make the article better understandable for the readers.

Section 1 and section 2 explain the background and pathogenesis of Apical Periodontitis. Starting from section 3 onward the author explained the possible/ currently available non-surgical treatment option for AP. The reviewer believes it is necessary to add some information related to the search criteria (optional as it is not a literature or systemic review), and insight on currently available articles (an elaborative form of table 1).

Section 3 onward the author explained several non-surgical treatment options for AP and the mechanism of action of such therapy. However, the information related to the success rate and limitations of these therapies is missing in the explanation. The reviewer believes this will add value to the article and will help the clinician choose the best treatment option to deal with individual cases. 

Reviewer 3 Report

Provide a clear and concise introduction to apical periodontitis, including its prevalence and clinical significance.   Use more recent and relevant references to support the information presented in the review.   Include a section on the current challenges in the diagnosis and treatment of apical periodontitis.   Use a more structured approach to present the advanced approaches for the treatment of apical periodontitis, such as grouping them according to their mechanism of action or clinical application.   Provide a more critical analysis of the studies reviewed, including a discussion of their limitations and potential biases.   Discuss the potential impact of these advanced approaches on patient outcomes, including the potential risks and benefits.   Conclude with a clear summary of the key findings and recommendations for future research.   Consider adding figures or tables to illustrate key points and data.   Use clear and concise language to improve readability and comprehension. Proofread the manuscript carefully for grammar, spelling, and formatting errors  

Reviewer 4 Report

In the Introduction topic, the subject of study is exposed, showing its importance, however the purpose of the article was not described at the end of the Introduction, requiring its inclusion.

Readjust the numbering of the article to 2.1, 2.2,2.3, 2.4 and 2.5, considering that Item 2 presents the 5 alternative therapies on page 2 of the manuscript. All alternative therapies addressed in the manuscript are new, a situation evidenced in several citations such as: more research and long-term evaluation are needed to verify these results” (p.4), “However, this mechanism requires further investigation” (p. 6).

The topic “Conclusions” presents a summary of recent advances in AP treatment, including the presence of a summary table.

Recommendations: - inclusion of the aim of the manuscript in the last paragraph of the Introduction; - change the subdivisions of the manuscript; - insert the discussion topic, where the contents present in the “Conclusions” topic can be reallocated to this part of the manuscript for critical analysis of the content; - construction of the CONCLUSION topic in response to the aim of the manuscript.

The manuscript proposes to address recent advances in therapy, however only therapeutic mechanisms are addressed. The manufactured products used in each of the alternative therapies addressed in the manuscript are not identified, nor there isn´t any mention of a suggestion of how they can be manufactured. These aspects of the manuscript compromise the applicability of the object of study and its clinical-scientific contribution to readers. However, in the Special Issue Information it is mentioned that “Their main objective is the application of engineering techniques for the fabrication of biological substitutes with the ability of totally or partially regenerating impaired organs or tissues.” Thus, the manuscript conforms to what is proposed in this Special Issue, being subject to publication after adaptations suggested by the reviewers.

Round 2

Reviewer 1 Report

Dear authors, I have no more comments. 

Reviewer 4 Report

All recommendations were made: - inclusion of the purpose of the manuscript in the last paragraph of the Introduction; - changes in manuscript subdivisions; - thread thread was built; the topic “Conclusions” was relocated to this part of the DISCUSSION manuscript for critical content analysis; - the topic CONCLUSION was constructed in response to the objective of the manuscript.